# Three-Dimensional Indoor Fire Evacuation Routing

**Yan Zhou [1], Yuling Pang [1,\*], Fen Chen [1] and Yeting Zhang [2]**

1   School of Resources and Environment, University of Electronic Science and Technology of China, Chengdu 611731, China; zhouyan_gis@uestc.edu.cn (Y.Z.); chenfen@uestc.edu.cn (F.C.)
2   State Key Laboratory of Information Engineering in Surveying, Mapping and Remote Sensing, Wuhan University, Wuhan 430072, China; zhangyeting@whu.edu.cn
\*   Correspondence: yulingpang@std.uestc.edu.cn

**Abstract:** Traditional indoor navigation algorithms generally only consider the geometrical information of indoor space. However, the environmental information and semantic parameters of a fire are also important for evacuation routing in the case of a fire. It is difficult for traditional indoor navigation algorithms to dynamically find an indoor path when a fire develops. To address this problem, we developed a multi-semantic constrained three-dimensional (3D) indoor fire evacuation routing method that considers multi-dimensional indoor fire scene-related semantics, such as path accessibility, path recognition degree, and fire parameters. Our method enhances the navigation semantics of indoor space by extending the fire-related components of indoor model based on IndoorGML and integrating location semantics of IndoorLocationGML. We also propose quantifiable indoor fire-oriented routing semantics and establish a navigation cost function that evaluates semantic changes during a fire. We designed an indoor routing algorithm with multiple semantic constraints based on the A* algorithm. The indoor routing results were analyzed and compared in simulation experiments. The experimental results showed that the proposed model can remove unusable nodes and edges from the obtained navigation path and provides a safer and more effective evacuation route than traditional algorithms.

**Keywords:** indoor routing; emergency evacuation; indoor 3D model; indoor navigation; fire semantics

## 1. Introduction

Ongoing urbanization has led to an increasing number of people working and living in large buildings [1]. Such environments are crowded, and a high number of casualties and property damage often occurs in the case of an emergency, such as a fire. The number of fires and direct property losses have been rising over the past 20 years [2]. Severe fires have raised extensive concerns in the government and the community. Indoor routing is critical for indoor fire emergency evacuation. Effective routing methods can help to reduce casualties and increase the escape rate of indoor people when a fire occurs.

Researchers have proposed some indoor routing methods. Some researchers mainly considered three-dimensional (3D) model semantic information [3–5], others focused on user information and its behavior semantics [6,7], and there are also some scholars focused on the environmental context and user preference semantics to provide personalized path planning services for users [8,9]. The main disadvantage of these methods is that they are designed for common circumstances, without taking special environments into consideration, such as a fire. These approaches calculated the shortest distance or time path as the optimal path, and in most cases, the shortest distance or the shortest time is their main path selection condition. However, shortest distance or time paths may not always provide safe paths in fire situations.

Fire-oriented indoor routing finds an optimal path based on an indoor model of a building in a fire scenario. Presently, indoor routing methods are presented based on diverse indoor models for indoor fire emergency evacuation (for detailed description, see Section 2). Some studies conducted path-finding based only on the geometric information of the indoor model. There are also some studies considered space semantics of the indoor model. However, most of the existing methods often ignored environmental semantics of fire scenarios. For example, when an indoor fire occurs, people must choose a path with minimal fire risk as the escape route. Considering only the shortest distance or route that takes the least time often fails when a fire spread. Factors such as path accessibility, path recognition degree, and fire spread must therefore also be considered.

In this paper, we model and quantify the semantics of fire environments, such as path accessibility, path recognition degree, and fire parameters, and integrate them into the indoor routing method. An extended indoor model for fire evacuation is designed, which is based on the framework of IndoorGML and integrated location semantics of IndoorLocationGML to enhance the navigation semantics of indoor space. We propose a multi-semantic constrained indoor routing method for indoor fire scenarios, which considers geometry and space semantics of indoor model, and environmental semantics of a fire scenario. The proposed method can provide users with intelligent, safe, and effective indoor 3D fire scenario routing information.

The remainder of this paper is organized as follows. Related works are discussed in Section 2. An extended indoor model for fire evacuation is designed in Section 3, and Section 4 presents an indoor routing method that accounts for the semantics of a fire scenario. Experiments and discussions are described in Section 5. The paper ends with conclusions in Section 6.

## 2. Related Works

Several previous studies presented some path-finding methods for indoor fire scenarios. Lee and Kwan developed a spatiotemporal optimal route algorithm based on the node-relation structure of indoor space, which was used to guide rescuers to quickly move from each building entrance to the disaster site inside the building [10]. Lu et al. proposed a fire evacuation method based on motion modes and cellular automata, which used the motion mode of intelligent mobile robots to simulate evacuation procedures during a fire [11]. Niu and Song presented a simulation model fusing indoor space and agent to simulate indoor evacuation process [12]. Atila et al. designed SmartEscape, a fire evacuation system, which calculated evacuation route using an artificial neural network [13]. Zhang and Wang developed a virtual reality system to simulate emergency evacuation in fires, and studied evacuation strategies in high buildings and human factors that could affect high-rise evacuation [14]. These studies planed evacuation paths based only on geometric information of indoor space, without taking space semantics of buildings into consideration. However, space semantics are importance in the context of indoor navigation, and make the path descriptions easily understandable.

Space semantics of buildings reflect the building subdivisions. Some models defined space semantics of buildings, such as the building information model (BIM) and IndoorGML. Some studies adopted BIM as indoor navigation model to find fire evacuation paths. Rueppel and Kai introduced a BIM-based indoor emergency navigation system to find the shortest way in a complex building [15,16]. Wang et al. combined BIM and virtual reality technologies to simulate virtual environment for fire emergency evacuation [17]. These studies designed BIM centered indoor navigation algorithms to support fire path finding, which integrated BIM to provide the geometric and semantic information of buildings as input to the indoor navigation algorithms. Though BIM has rich geometric and semantic information of buildings, its time-consuming to process the amount of building geometries and semantics. It was observed that the building geometry processing could decrease the efficiency of BIM-based indoor navigation algorithms [18].

The Open Geospatial Consortium (OGC) proposed IndoorGML, which defined generic semantics of indoor space subdivision for navigation purposes, such as navigable space and non-navigable space. Compared to BIM, IndoorGML provides a concise expression of indoor spaces semantics

and a lightweight indoor navigation model [19]. Xiong et al. proposed a dynamic indoor field model for emergency evacuation simulation, which expanded IndoorGML by defining three core objects (indoor space, indoor emergency grid, and grid unit) [20]. Zhu et al. introduced a Chinese national standard of indoor navigation model: Indoor Multi-Dimensional Location Geography Markup Language (IndoorLocationGML), which focused on the descriptions of indoor locations and provided a framework of indoor location services [21]. Liu et al. took an initiative in attempting to integrate IndoorGML and IndoorLocationGML and presented the possibility to combine two indoor-related standards for indoor applications [22]. Alattas et al. presented a combined LADM-IndoorGML model, which combined use of the indoor IndoorGML and LADM models to support indoor navigation using access rights to spaces [23,24]. Moreover, an innovative indoor routing algorithm on logical network was designed, which computed a logical path based only on space semantics, not geometric information of building [25]. Compared to the indoor navigation studies based only on the geometrical information of indoor space, these researches took space semantics of buildings into consideration. However, the environmental information and semantic parameters of a fire are also important for evacuation routing in the case of a fire. Most of the existing methods often ignore environmental semantics of fire scenarios, such as path accessibility, path recognition degree, and fire spread, etc.

In the next sections, we propose an extended indoor model for fire evacuation and a fire-oriented indoor routing algorithm based on this model.

## 3. An Extended Indoor Model for Fire Evacuation

In this section, we analyze the characteristics and influencing factors of indoor building fires and extend the indoor fire-related components, then integrate location semantics of IndoorLocationGML into an indoor model to enhance navigation semantics of indoor space for fire evacuation.

### 3.1. Indoor 3D Space Model

The OGC first drafted an open data model for storing and exchanging virtual 3D urban models—CityGML [19]. However, CityGML's division of indoor building components is not clear and cannot determine the reachability of indoor spaces. The OGC therefore specifically proposed IndoorGML for indoor spaces, which focuses on representing the semantic information of indoor space and indoor navigation networks [19]. IndoorGML provides a general framework and semantic descriptions for an indoor navigation. However, IndoorGML only focuses on modelling indoor space for lightweight navigation purpose, which cannot fully meet the needs of ubiquitous indoor location services [21]. Industry Foundation Classes (IFC) are the actual data standard for international construction industry [26]. BIM is a data model established on the basis of IFC for building a field with rich geometric and semantic information. Many scholars have conducted researches on BIM-based indoor emergency navigation [15–18]. However, indoor emergency navigation is time-sensitive for fire scenarios. When indoor routing algorithms is designed to be efficient, it relies on the efficiency of processing building geometries. As the amount of BIM data is usually very large especially for complex buildings, the BIM geometry processing takes up most of the total computational time [18]. These BIM-based routing studies did not focus on improving efficiency of processing building geometries. At the end of 2017, the China National Standards Committee officially released the first national standard for indoor location services: IndoorLocationGML, which defines the indoor multi-dimensional location information model required for indoor positioning and navigation. IndoorLocationGML focuses on the descriptions of indoor locations (e.g., accurate and relative locations of indoor space, semantics, and topology of locations, etc.), which aims to meet ubiquitous indoor location service requirements [21,27]. Liu et al. designed a joint model by integrating IndoorGML and IndoorLocationGML [22]. IndoorGML, IndoorLocationGML, and the joint model that combines them are all designed to meet the general requirements of indoor navigation and location service, without taking fire scenario into consideration. Due to the lack of semantic expression for indoor

fire-related elements (e.g., fire doors, firefighting facilities, evacuation indicating lamps, etc.), they cannot meet the indoor emergency navigation requirement in fire situations.

To address this issue, we extend the indoor fire-related components based on IndoorGML and combined it with location semantics of IndoorLocationGML to build a new indoor space model for fire evacuation. We further study the routing method for indoor fires based on this model.

### 3.2. Characteristics of Indoor Fires in Buildings

There are some factors that can affect emergency evacuation route selection in the event of an indoor fire. In this section, we analyze the characteristics of indoor fires and model the requirements for indoor fire scenarios.

1.  Fire spreads fast. In the event of a fire, smoke can quickly spread in stairwells, elevators, and other vertical corridors. At this time, fire doors and windows installed in the stairwell or vertical passageway are important fire prevention measures that can prevent the flow of combustion smoke during the fire. Identifying the spatial location of these building facilities can prevent the spread of fire to a certain extent and ensure safe evacuation.
2.  High hazard. The interior of modern buildings is generally densely populated and complex in spatial structure. Some obstacles are flammable and explosive, which will become very dangerous in the case of a fire. If people are not familiar with the distribution of indoor exits or if emergency evacuation signs are absent in the building, the evacuation difficulty will increase.
3.  Difficult to fight fire. Fires are known to be quite difficult to fight from the outside. People mainly use indoor fire protection facilities to extinguish fires [28]. If the locations of the firefighting facilities inside a building are unclear, the fire will be uncontrollable.

### 3.3. Influencing Factors of Routing in An Indoor Fire

Due to the characteristics of indoor fires and evacuation difficulty in high buildings, it is necessary to consider various important factors that affect path planning in indoor fire scenarios. In the following, we analyze several factors in an indoor fire scene.

- Evacuation components
- Exit and route selection
- Path security

Modern buildings have complex structures and a large number of evacuation channels. If daily management is inadequate, the security evacuation channels can be blocked, such as locked safety doors, safety exits, or the accumulation of items in stairwells. When a fire occurs, the locations and states of these safety evacuation components may be unclear, which is an important reason for many casualties.

Emergency evacuation exits and route selection are important factors that affect path planning under certain circumstances. For instance, in extreme cases of escape from fire, windows on the lower floors and fire ladders can be used as an anchor space and evacuation exits to avoid congestion, stomping, and jumping to achieve safe evacuation. The correct selection of emergency evacuation exits can therefore increase the effectiveness of emergency evacuation.

As mentioned, indoor fires are characterized by a large number of hazards. Flammable and explosive objects can be situated in dangerous locations, which then affects the choice of evacuation path in the case of an indoor fire. Additionally, a large amount of smoke and dust in passageways during a fire greatly reduce the visibility of trapped people, which affects the evacuation speed and safety. Smoke prevention, exhaust devices, and evacuation-indicating lighting that guide evacuees in low visibility are therefore very important for the evacuation of people during a fire.

### 3.4. Extension of Components for Indoor Fire Scenarios

Traditional indoor models focus on the expression of indoor structure and routing under normal circumstances. However, some overlooked indoor components can play a key role in the case of fire emergency. There is presently no unified semantic description of indoor space for different applications. To meet indoor fire evacuation requirements, we therefore refer to IndoorGML [19] and add fire-related components to propose an extended indoor space description model for indoor fire scenarios. The indoor semantic structure description diagram of this model is shown in Figure 1.

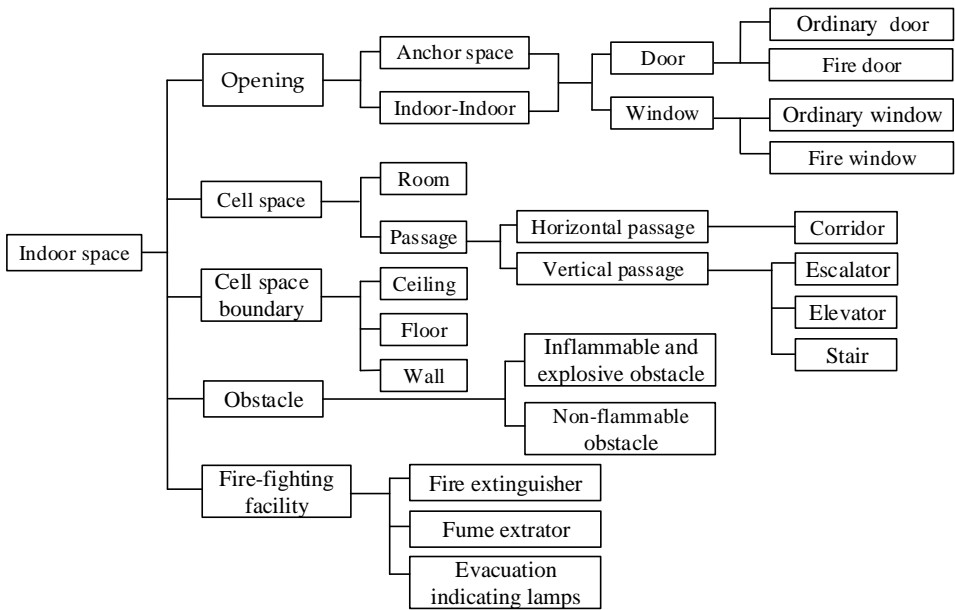

**Figure 1.** Indoor semantic structure for indoor fire scenarios.

This model mainly extends the following four aspects:

1. In terms of indoor components that affect evacuation in indoor fire scenarios, we extend the fireproof door and window components and firefighting facility components. Fireproof doors and windows are building components that can meet the requirements of fire resistance stability, integrity, and heat insulation within a certain time. These include fire-resistant objects with certain fire resistance, which are arranged in a fire-resistant partition room, evacuation stairwells, and vertical shafts. For firefighting facilities, we extend the number of fire extinguishers, smoke prevention and exhaust devices, and fire diversion signs.

2. In terms of the emergency exit selection for indoor fire scenarios, we extend the window exit components in the anchor space. The anchor space refers to the exits and entrances that connect the building interior and exterior. Extending the number of window exits in the anchor space can help users to choose a low-floor window or fire ladder to escape in certain circumstances.

3. In terms of the path selection of an indoor fire scenario, we extend the elevator, escalator, and staircase components in the vertical passageway to distinguish the vertical reachable path in a fire. Elevators do not work during fires due to the cessation of the power supply. Escalators are generally located in relatively open halls and can objectively play a role in the early fire period. However, for safety reasons, escalators cannot be used as evacuation facilities in accordance with national regulations. Stairs are therefore a relatively safe selection in a vertical passageway.

4. In terms of the safety of evacuation in an indoor fire scenario, we extend the flammable and explosive objects and non-flammable object components as obstacles. In the general indoor navigation process, obstacles affect the path planning and complicate the navigation process. The IndoorGML standard classifies obstacles as non-navigational space and does not consider

the obstacle characteristics in detail. However, the flammable and explosive characteristics of obstacles must be considered to increase safety during an emergency fire evacuation.

### 3.5. Location Definition Based on Indoor Location GML

The IndoorLocationGML standard contains extensive location information, of which the abstract indoor location class is an important part. As the base class of some other indoor location type, it is classified into two parts: indoor absolute location and indoor relative location. The indoor absolute location expresses a point or interval in indoor space by defining the geometric coordinates in a given spatial reference system. The geometric coordinates are defined in a given spatial reference system and described with a geometric coordinate class. The indoor relative position is defined as the position of an object relative to other indoor reference objects. It is expressed as the description of a certain position in indoor space through the relative relationship with other positions in the same coordinate reference system, including the relative geometric location and semantic location, from the geometric and semantic perspectives, respectively.

We propose an extended indoor model for indoor fire evacuation, which added indoor fire-related components based on the framework of IndoorGML and integrate location semantics of IndoorLocationGML into indoor model to enhance navigation semantics of indoor space for fire evacuation. The Unified Modeling Language (UML) diagram of our proposed indoor model is shown in Figure 2.

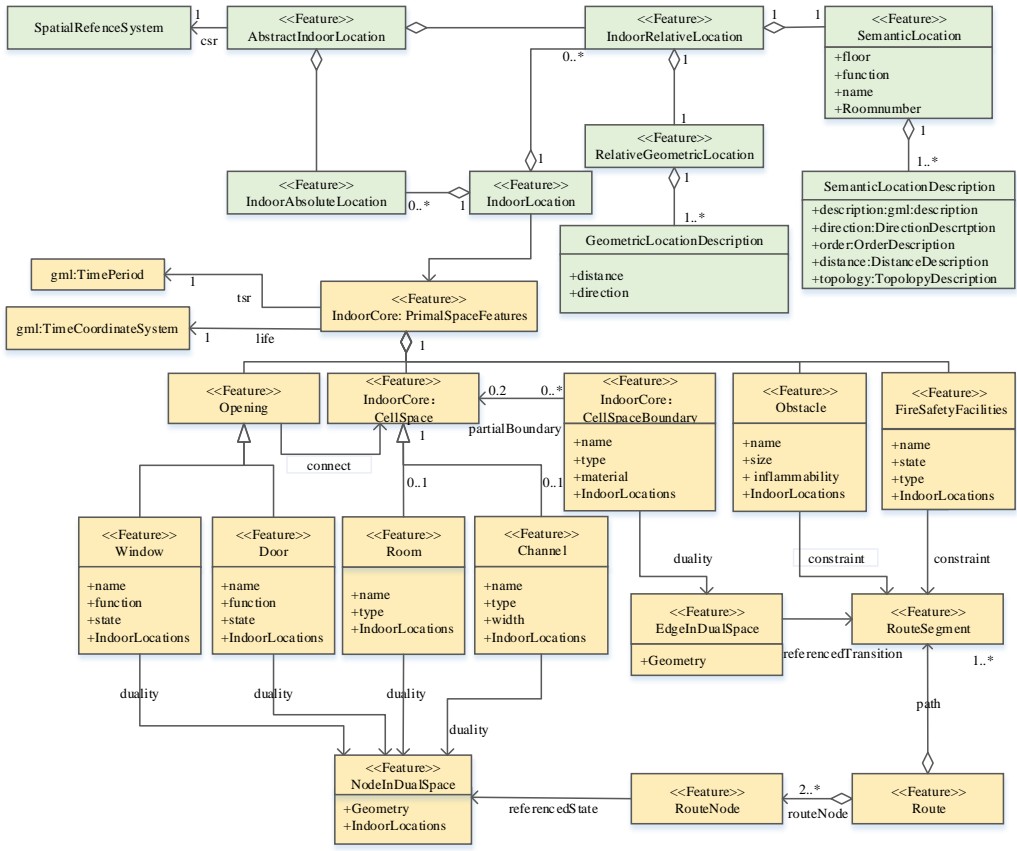

**Figure 2.** The Unified Modeling Language (UML) diagram of indoor model for fire evacuation.

PrimalSpaceFeatures Class. The PrimalSpaceFeatures class is the base class for any other indoor space object, which is based on IndoorGML [19]. It is associated with the time reference system and life cycle classes. The indoor time reference system [29] is expressed in GML: TimeCoordinateSystem. It defines a continuous time interval scale in the form of a single time interval, which represents the

tense of the indoor space characteristics. The position of the gml: TimePeriod, is described by the corresponding moment at the start and end.

1.  Opening. The Opening class is inherited from the PrimalSpaceFeatures class and includes the Door class and Window class, which contain four attributes: name, function, accessibility, and IndoorLocations. The name is an xs-type string, namely, a string consisting of numbers and letters. The function is defined by the WindowFunctionType and DoorFunctionType, which include fire doors and windows, anchor doors and windows, and ordinary doors and windows. The state indicates reachability through a Boolean identifier, where 0 indicates that a door or window is impassable and 1 indicates that a door or window is passable. IndoorLocations is associated with indoor location descriptions defined by IndoorLocationGML [27].
2.  CellSpace. A CellSpace is the IndoorGML core module, which is a semantic class corresponding to one space object in Euclidean space [19]. The CellSpace is inherited from the PrimalSpaceFeatures class, which includes the Room class and Channel class. The Room class contains two attributes: name and IndoorLocations. The Channel class contains four attributes: name, type, width, and IndoorLocations.
3.  CellSpaceBoundary. A CellSpaceBoundary is also the IndoorGML core module, which is used to semantically describe the boundary of a space object [19]. The CellSpaceBoundary is also inherited from the PrimalSpaceFeatures class and is associated with CellSpace.
4.  Obstacle. The obstacle is an extension class that includes four attributes: name, size, inflammability, and IndoorLocations.
5.  FireSafetyFacility. The FireSafetyFacility is an extension class that includes four attributes: name, state, type, and IndoorLocations. The types of fire facilities include FireExtinguisher, FumeExtrator, and EvacuationIndicatingLamps.
6.  Dual Space. Through the dual transformation, a k-dimensional object in N-dimensional space can be transformed into a (N-k) dimensional object in dual space. This means that in the corresponding dual space, a 3D cell space (e.g., room) is converted into a zero-dimensional node. A 2D boundary surface (e.g., wall and door) between two units is converted into a 1D edge. In this way, the original space is transformed into a node graph of dual space to express the topological relationship of the indoor road network. Figure 3 shows the example of dual transformation.
7.  Indoor road network. Based on the room–room model, interior rooms, stairs, elevators, escalators, doors, and windows are implemented as nodes in dual space. We add additional nodes at the connection points of the channel components, as well as door, window, and channel components. The cell space boundary of the corridor and vertical channel is implemented as an edge in dual space. We construct indoor path points and path segments using obstacles and firefighting facilities as semantic constraints. We then obtain a fire-oriented indoor 3D road network model.

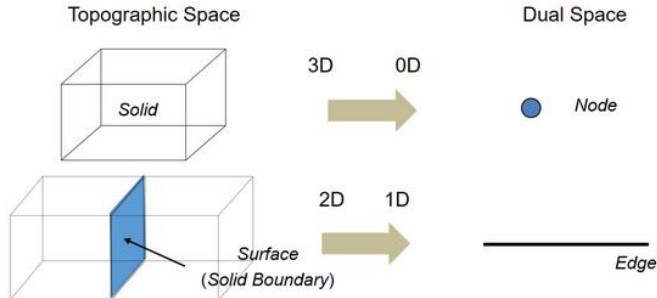

**Figure 3.** Example of dual transformation [30].

## 4. A Routing Method for Indoor Fire Scenarios

### 4.1. Semantic Expression of Indoor Routing

In the event of a fire, indoor routing must be accessible, efficient, and safe. We therefore consider the path accessibility of a fire scenario, path recognition degree, and fire parameters as the main influencing factors for fire-oriented routing to express the fire-oriented indoor routing semantics.

#### 4.1.1. Path Accessibility of a Fire Scenario

Path accessibility is an indicator of how easy it is to move between nodes in a road network. Generally, some nodes are objectively unreachable (e.g., locked doors and windows). However, during a fire, the accessibility state of doors and windows may change, and escalators and elevators cannot be used as evacuation routes. The accessibility state of indoor components therefore directly affects the evacuation speed and evacuation path choice and reduces the capacity of the evacuation passageway. We divide obstacles into non-flammable obstacles and flammable explosive obstacles. The presence of non-flammable obstacles is considered to change the available paths when people are evacuating. The path accessibility in a fire scenario can therefore be described as

$$path\_accessibility = \{vetax\_state,\ number\_unfl,\ number\_fl\} \tag{1}$$

where *vetax_state* is the state of the road network node and is expressed by two types of road network node states, reachable and unreachable. If the node state is unreachable, the node cannot be used during an evacuation. Table 1 presents the states of doors, windows, and vertical access stairs, elevator escalator nodes in the fire scene. *number_unfl* and *number_fl* are the numbers of non-flammable and flammable explosive obstacles in the path, respectively.

**Table 1.** Road network node state in an indoor fire scenario.

| Vetax | Vetax_State | Conditions |
|---|---|---|
| Ordinary door | Reachable<br>Unreachable | Unlocked<br>Locked |
| Fire door | Reachable<br>Unreachable | Open<br>Closed |
| Ordinary window | Reachable<br>Unreachable | Unlocked and the size and height are sufficient for people to pass<br>Locked and the size and height are insufficient for people to pass |
| Fire window | Reachable<br>Unreachable | Open<br>Closed |
| Anchor space door | Reachable<br>Unreachable | Unlocked<br>Locked |
| Anchor space window | Reachable<br>Unreachable | Unlocked and located on the first floor<br>Locked and located on the second floor and above |
| Stairs | Reachable<br>Unreachable | Unlocked<br>Locked |
| Elevator | unreachable | fire occurs |
| Escalator | unreachable | fire occurs |

#### 4.1.2. Path Recognition Degree of a Fire Scenario

A large amount of dense smoke generated by a building fire combustion greatly reduces the visibility range and cause problems with evacuation. At this point, evacuation-indicating lighting is a key element of indoor navigation routing, as it can reduce the cognitive burden during the evacuation process and enhance navigation confidence. Such lighting also has a calming effect and ensures that correct routing decisions are made [31]. However, there is often a serious lag in the maintenance and overhaul of evacuation-indicating lighting, making it impossible for failures in the emergency evacuation indicators to be detected in time. The failure of evacuation-indicating lamps in the safe passageway creates blind spots and affects the identification of an escape path in a fire scenario.

The number of working evacuation-indicating lamps is often used to characterize the path recognition degree of a fire scenario. Its semantic expression can be described as

$$path\_recognizability = \{number\_lamp\} \tag{2}$$

where *number_lamp* is the number of effective evacuation indicating lamps along the evacuation path.

### 4.1.3. Fire Parameters

An indoor fire is a complex disaster coupled by multiple factors. Previous studies have shown that fire temperature and visibility are more important to health and evacuation than the gas concentration in smoke. The semantics of indoor routing therefore considers fire temperature and visibility.

With increasing thermal energy radiation, the high temperatures generated by building fires cause people to have a high body temperature, surface burns, and respiratory tract burns, which threaten their safety [32]. Fire temperature is therefore an important indicator to measure the degree of danger. We describe the semantics of fire temperature as

$$fire\_temperature = \{T,\ t - range,\ Tt\} \tag{3}$$

where $T$ is the temperature, $t - range$ is the set of nodes affected by the temperature, and $Tt$ is the time of the current fire temperature. Humans can only endure temperatures in the range of 42–50 °C for a very short time. Without protective clothing, humans cannot move in high-temperature air at 50 °C [33]. As shown in Table 2, we divide the temperature parameter into four intervals according to the danger level, and the $t - range$ is described as

$$t - range = (v_1,\ v_2, \cdots,\ v_n) \tag{4}$$

Visibility generally refers to the longest distance that a person can see an object. Smoke from a building fire has the characteristics of being lightproof, and a visibility indicating lamp is an important factor in the safe evacuation of people in a smoke environment. We describe the semantics of smoke visibility as

$$smoke\_visibility = \{V,\ v - range,\ Tv\} \tag{5}$$

where $V$ is the visibility parameter and $v - range$ is the node-set affected by the visibility. According to statistical research in Australia's "Fire Engineer's Guide," people in a large space need to be able to see farther than people in a small space to find the evacuation direction, and therefore require better visibility [34]. Table 3 lists the specific visibility thresholds. For general buildings, the length of the evacuation corridor is relatively long, and it is more difficult for people to find the evacuation direction and path in time during a fire. We define the visibility fire parameter range according to the influence of visibility on path finding, as shown in Table 4. $Tv$ represents the time of the current fire visibility.

**Table 2.** Fire temperature parameter range.

| Temperature Range (°C) | <42 | 42–50 | 50–80 | >80 |
|---|---|---|---|---|
| Levels of danger | Safe | Potential danger | Danger | High danger |

**Table 3.** Visibility threshold for large and small spaces.

| Location | Small Space | Large Space |
|---|---|---|
| Visibility threshold (m) | 5 | 10 |

| Visibility Interval (m) | >10 | 5–10 | <5 |
|---|---|---|---|
| Level of danger | Safe | Potential danger | High danger |

### 4.2. 3D Navigation Cost of Indoor Fire Scenarios

Based on the semantic expression of 3D routing for indoor fire scenarios, we construct corresponding path-cost functions that consider path accessibility in a fire scenario, path recognition degree, and fire parameters.

### 4.2.1. Path Accessibility Cost Function

Path accessibility in fire scenarios is an important indicator of the difficulty of moving between nodes in a road network and is also an important decision factor for path selection in fire situations. The path accessibility of a fire scene is constrained by three factors: node state, number of non-flammable obstacles, and number of flammable and explosive obstacles. The path accessibility cost function is defined as

$$f_{acc}(v_i, v_j) = \frac{f_{unfl}(v_i, v_j)}{f_{state}(v_i) \times f_{state}(v_j) \times f_{fl}(v_i, v_j)} \tag{6}$$

where $f_{acc}(v_i, v_j)$ is the cost function of the path accessibility from node $v_i$ to node $v_j$. $f_{state}(v_i)$ and $f_{state}(v_j)$ are the accessibility state of node $v_i$ and node $v_j$ in a fire scenario. The accessibility state can be judged using Table 1 and the reachable state of node $v_i$ is expressed as follows, and likewise for $f_{state}(v_j)$ of node $v_j$.

$$f_{state}(v_i) = \begin{cases} 1, & reachable \\ 0, & unreachable \end{cases} \tag{7}$$

where $f_{unfl}(v_i, v_j)$ is the path accessibility influence cost function of non-flammable obstacles from node $v_i$ to node $v_j$ in a fire scenario and is expressed by the number of non-flammable obstacles as

$$f_{unfl}(v_i, v_j) = number\_unfl \tag{8}$$

where $f_{fl}(v_i, v_j)$ is the path accessibility influence cost function of flammable and explosive obstacles from node $v_i$ to node $v_j$ in a fire scenario. Due to the extremely high risk of flammable and explosive obstacles, we define the section with flammable and explosive obstacles as unreachable and its function is expressed as

$$f_{fl}(v_i, v_j) = \begin{cases} 1, & there\ are\ no\ flammable\ or\ explosive\ obstacles \\ 0, & there\ are\ flammable\ or\ explosive\ obstacles \end{cases} \tag{9}$$

### 4.2.2. Path Recognition Degree Cost Function

The path recognition degree in a fire scenario indicates the difficulty that people have in perceiving, recognizing, and executing route navigation instructions during an evacuation, and is an important factor that affects the evacuation safety and efficiency. A large amount of thick smoke reduces visibility in the internal building space, and emergency evacuation indicating lighting that guides the evacuation greatly improves the evacuation efficiency. However, in the case of inadequate maintenance and repair of the emergency evacuation indicators, there is usually a lag phenomenon, which causes indicator failure. Once a fire or other emergency occurs, this presents a serious problem for path recognition.

In the case of a fire, the number of evacuation indicators that work is therefore normally used to characterize the path recognition degree in fire scenarios. Its semantic expression can be described by

$$f_{rec}(v_i, v_j) = \frac{d(v_i, v_j)}{number\_lamp + 1} \tag{10}$$

The path recognition degree of fire scenario $f_{rec}(v_i, v_j)$ is defined by Equation (10), where *number_lamp* is the number of effective evacuation indicating lamps from node $v_i$ to node $v_j$, and $d(v_i, v_j)$ is the indoor relative distance between $v_i$ and $v_j$, defined as the Euclidean distance. If the geometric absolute locations of nodes $v_i$ and $v_j$ in a 3D expression model for indoor fire scenarios are $(x_i, y_i, z_i)$ and $(x_j, y_j, z_j)$, the indoor relative location distance between these two nodes is expressed as

$$d(v_i, v_j) = \sqrt{(x_i - x_j)^2 + (y_i - y_j)^2 + (z_i - z_j)^2} \tag{11}$$

### 4.2.3. Cost Function of Indoor Fire Parameters

When an indoor fire occurs, its magnitude directly affects path selection and personal safety. Fire parameters are therefore important factors that characterize the magnitude of an indoor fire. Temperature and visibility, which are important for evacuation and people's health, are selected as indoor fire parameters. The cost function is defined as

$$f_{fire}(v_i, v_j) = \alpha(T, V) \times d(v_i, v_j) \tag{12}$$

where $d(v_i, v_j)$ is the distance between nodes $v_i$ and $v_j$ in the 3D expression model of indoor fire scenarios, and $\alpha(T, V)$ is a coefficient related to fire temperature and visibility defined according to the division of the fire temperature range in Table 2 and the visibility in Table 3 as

$$\alpha(T, V) = \begin{cases} 1.0, & T < 42\,°\text{C},\ V > 10 \\ \frac{T}{42} + \frac{5}{V}, & 42\,°\text{C} \leq T \leq 50\,°\text{C},\ 5\,\text{m} \leq V \leq 10\,\text{m} \\ \infty, & \text{else} \end{cases} \tag{13}$$

### 4.2.4. 3D Navigation Cost Function for Indoor Fire Scenarios

According to the cost function of path accessibility, path recognition degree, and the fire parameters, we quantify the cost function of indoor 3D navigation for a fire scenario as

$$G = \omega_a f_{acc}(v_i, v_j) + \omega_r f_{rec}(v_i, v_j) + \omega_f f_{fire}(v_i, v_j) \tag{14}$$

where $f_{acc}(v_i, v_j)$ is the cost function of path accessibility in a fire scenario from node $v_i$ to node $v_j$, $f_{rec}(v_i, v_j)$ is the cost function of the path recognition degree in a fire scenario, and $f_{fire}(v_i, v_j)$ is the cost function of the fire parameters. According to Equations (6), (10), and (12), we can expand Equation (14) to obtain:

$$G = \omega_a \frac{f_{unfl}(v_i, v_j)}{f_{state}(v_i) \times f_{state}(v_j) \times f_{fl}(v_i, v_j)} + \omega_r \frac{d(v_i, v_j)}{number\_lamp + 1} + \omega_f \left[ \alpha(T, V) \times d(v_i, v_j) \right] \tag{15}$$

where $\omega_a$, $\omega_r$, and $\omega_f$ are weight coefficients. Weights can be allocated according to user needs, and the following relationships are satisfied:

$$\sum \omega_i = 1, \omega_i \in (0, 1), i \in \{a, r, f\} \tag{16}$$

In the multi-semantic constrained indoor routing simulation experiment, the weights are set to $\omega_a = 0.35$, $\omega_r = 0.3$, and $\omega_f = 0.35$.

### 4.3. 3D Routing Algorithm for Indoor Fire Scenarios

The essence of indoor optimal path planning is to find the path with the smallest cost between the starting and end points. Classic shortest route-planning algorithms mainly include those of Dijkstra [35,36], A* [37,38], Bellman-Ford [39], Floyd [40,41], and Shortest Path Faster Algorithm(SPFA) [42,43]. As a heuristic search method, A* can quickly respond to environmental information. It is an effective search method for solving the shortest path and is a commonly used heuristic algorithm for many routing applications. We therefore improve the A* algorithm and propose a 3D routing algorithm for indoor fire scenarios. The algorithm is transformed to solve the optimal routing problem with the least cost in an indoor navigation road network with multi-factor fire environment semantic constraints.

#### 4.3.1. A* Algorithm

The A* algorithm is a heuristic path search algorithm proposed by Hart et al. in the late 1960s [44]. As a heuristic search algorithm, A* is an informed search algorithm that judges a path based on a weighted graph; starting from a specific starting node of a graph, given the path cost of the target node, to find the path with the smallest cost. The principle is based on the selected nodes: a designed evaluation function $F(n) = G(n) + H(n)$ guides the next expansion of the node, where $F(n)$ estimates the cost of the known starting node through node $n$ to reach the target node. $G(n)$ defines the actual cost from the known starting node to the current node $n$, and $H(n)$ is the estimated cost of the optimal path from the current node $n$ to the target node. The cost of each node can be calculated according to the evaluation function, and each node that can be reached in the next step is evaluated by the heuristic function. In the search, the node with the smallest value is found and the search is continued until the target node is reached. The A* algorithm is simple and intuitive. However, in large engineering projects, the A* algorithm often needs to search all nodes to find the shortest path. Choosing an appropriate evaluation function can therefore make the routing algorithm more efficient and easy to implement.

The selection of an evaluation function directly affects the search efficiency of the heuristic search algorithm. A higher performance of the selected evaluation function will lead to a lower number of nodes being searched by the algorithm, less calculation time, and a higher search efficiency for finding the optimal path [45]. According to the analysis in Section 4.2, $G(n)$ can be calculated in the evaluation function $F(n) = G(n) + H(n)$ according to the navigation traffic cost function proposed here. The current commonly used heuristic function $H(n)$ is mainly calculated by the distance function. In this study, we use the Euclidean distance as the heuristic function of the A* algorithm.

#### 4.3.2. Multi-Semantic Constrained Indoor Routing Algorithm

Based on the A* algorithm, the cost function of the indoor routing algorithm with multiple semantic constraints is set as

$$F(n) = G(n) + H(n) \tag{17}$$

where $G(n)$ represents the cost function from the start node to the node $n$, and $G(n)$ is calculated by Equation (14). The evaluation function $H(n)$ represents the Euclidean distance from node $n$ to the target node. The proposed indoor evacuation path finding algorithm constrained by multi-semantic flow is shown in Figure 4.

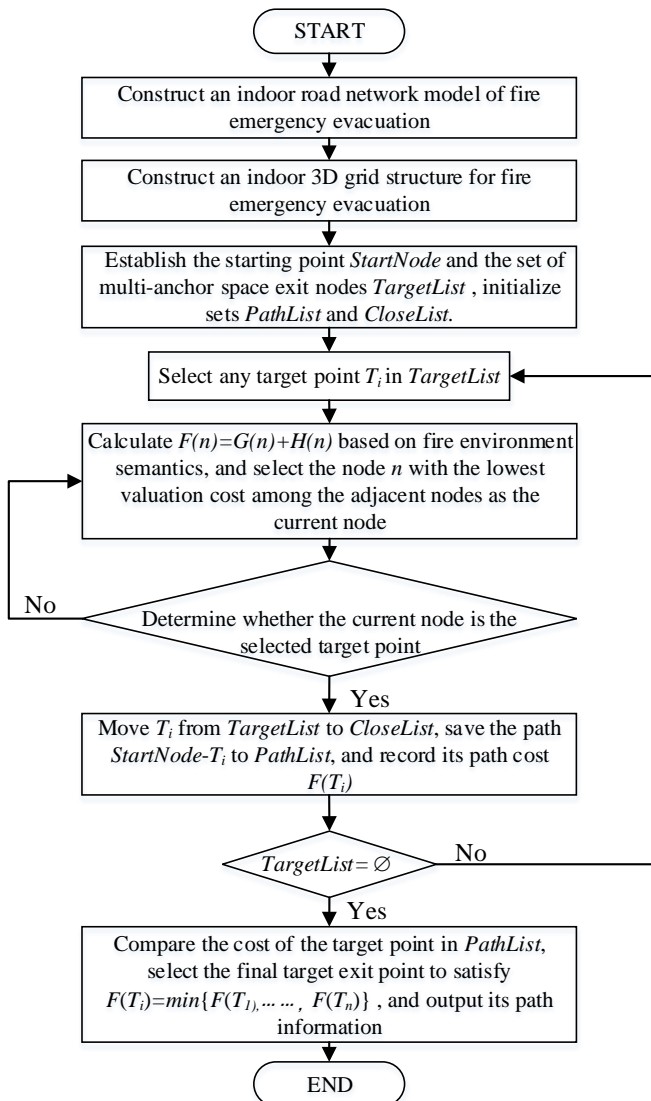

**Figure 4.** Flow chart of indoor routing algorithm with multiple semantic constraints.

1. Construct a 3D grid structure of vertical passageways and connect each floor by a vertical passageway to form an indoor 3D grid structure for indoor fire scenarios. Elevators and escalators are inaccessible during the emergency evacuation process; thus, people need to choose stairs as the evacuation route.

2. Establish a starting point and set of multi-anchor space exit nodes. Determine the location of the starting point of the path search for a fire scenario. Select the nearest node in the indoor road network model as the starting point *StartNode*. Doors and windows are used as anchor space exit nodes to satisfy the emergency evacuation requirements. Given the risk of escape from an anchor space window exit on the second floor and above, only the first-floor door and window exits in the anchor space are added to the multi-anchor space exit point set *TargetList* $=\{T_1, T_2, \ldots\ldots, T_n,\}$. The sets *CloseList* and *PathList* are initialized to store the traversed target point and navigation path from the starting point to the target point.

3. Calculate the evaluation function $F(n) = G(n) + H(n)$ and select the node with the lowest valuation cost between adjacent nodes as the current node. Determine the indoor 3D navigation cost function for fire scenarios $G(n)$ accounting for the semantics of indoor routing proposed in this paper. Select the target point $T_i$ in *TargetList* and calculate the Euclidean distance from all adjacent nodes at the current starting point to the target point $T_i$ as the heuristic function $H(n)$.

Calculate the evaluation function $F(n) = G(n) + H(n)$ and select the path adjacent point with the lowest evaluation function value as the current node.

4.  Determine whether the current node is the target point $T_i$ selected in step (3). If yes, move $T_i$ from the multi-anchor space exit set *TargetList* to the traversed anchor space exit set *CloseList*, add the walk path from its starting point to the current point in the set *PathList*, and save its navigation cost. Otherwise, return to step (3) and calculate and compare the value of the node valuation function adjacent to the current node.

5.  Determine whether the set is empty. If the set is not empty, insert a new node in the set as the target point and return to step (3). If the set is empty, this means that all target points in the multi-anchor space exit set have been traversed. Compare the navigation costs of these feasible paths to determine the optimal path. This is satisfied by $F(T_i) = \min\{F(T_1), \ldots\ldots, F(T_n)\}$. Output the selected path information.

6.  End of the algorithm.

## 5. Experiments and Discussions

### 5.1. Construction of An Indoor 3D Expression Model for A Fire Scenario

We modeled a building according to the 3D expression model for indoor fire scenarios established in this paper. The mode is shown in Figure 5. The 3D building model was then abstracted into a road network model composed of nodes and edges. Among them, rooms, doors, and windows were abstracted as geometric centers expressed as nodes; corridors, elevators, stairs, and other channels were abstracted as edges in the form of center lines. The connection points of doors, windows, vertical channels, and corridors were added as additional nodes. The node-to-node and node-to-edge establish the node-to-edge connection based on the actual path connectivity to achieve indoor road network path connectivity. Figure 6 shows a concise example for subdivision of single-layer indoor space based on our model, which provides navigable subspaces based on meshes.

The building model was constructed based on the indoor 3D expression model for a fire scenario proposed in this paper. The abstract indoor 3D road network model is shown in Figure 7. Black nodes represent the abstract nodes of rooms, elevators, and stairs, red nodes represent corresponding doors, blue nodes represent the corresponding windows, and pink nodes represent the additional nodes added in the corridor position.

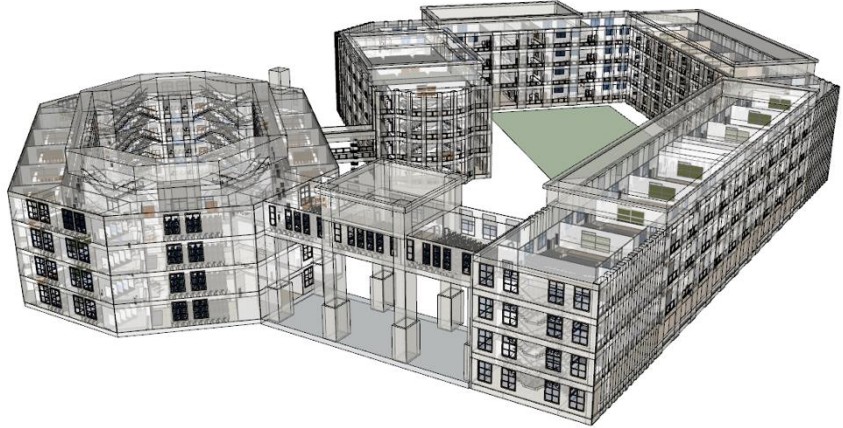

**Figure 5.** 3D building model.

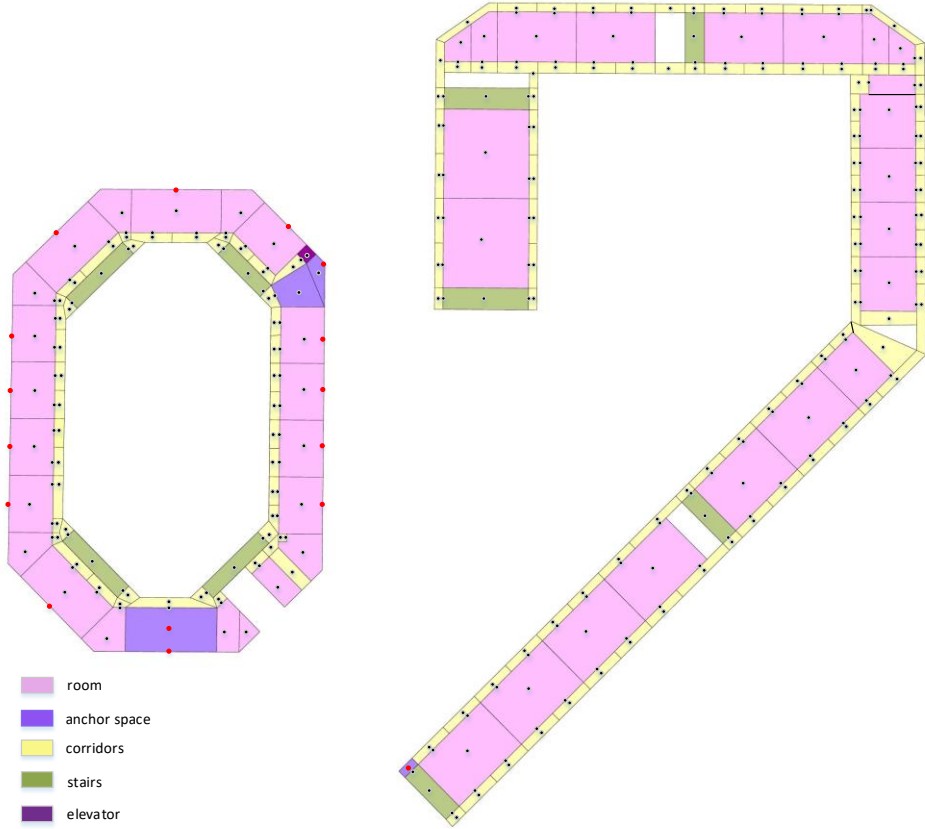

room
anchor space
corridors
stairs
elevator

**Figure 6.** A navigation grids diagram of single-layer indoor space.

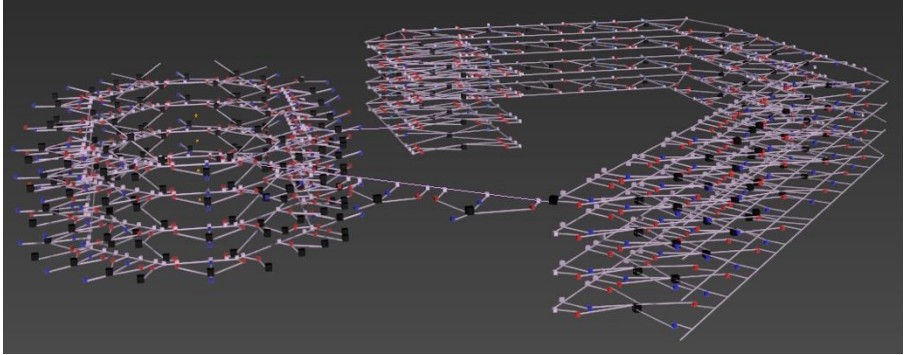

**Figure 7.** Indoor 3D road network model.

*5.2. Fire Simulation*

For the fire simulation, we used the Fire Dynamics Simulator (FDS) developed by the National Institute of Standards and Technology, which is widely used in computational fluid mechanics simulation research on fluid flow under fire conditions. The FDS uses a mixed-scale model to simulate the combustion process and the Navier-Stokes equation to calculate various combustion parameters in a fire. This was used to analyze the movement process of smoke and heat in a fire.

The established model was first imported into the FDS. The measuring point was set in the building according to the node position in the indoor road network model for the fire scenario. The thermocouple stylus was used to monitor the visibility and temperature data when a fire occurs and provide data support for the indoor routing algorithm.

The FDS smoke and temperature simulation results were displayed in SmokeView, as shown in Figure 8. After 60 s, black smoke appears in the building and a small amount diffuses while the

temperature slowly rises to a maximum value of 50 °C. At this time, visibility is the main factor that affects evacuation. After 120 s, the temperature in some areas reaches a dangerous level and people should avoid these areas and choose other safe areas to escape. After 240 s, although the temperature does not reach a level that can burn people, the building is filled by a large amount of black smoke. Due to its strong lightproof property, visibility and the path recognition in the fire scenario are reduced, which increases the difficulty of evacuation. After 600 s, a large area of high-temperature smoke appears at the fire source, reaching a dangerous level, and people must urgently find a safe path to evacuate.

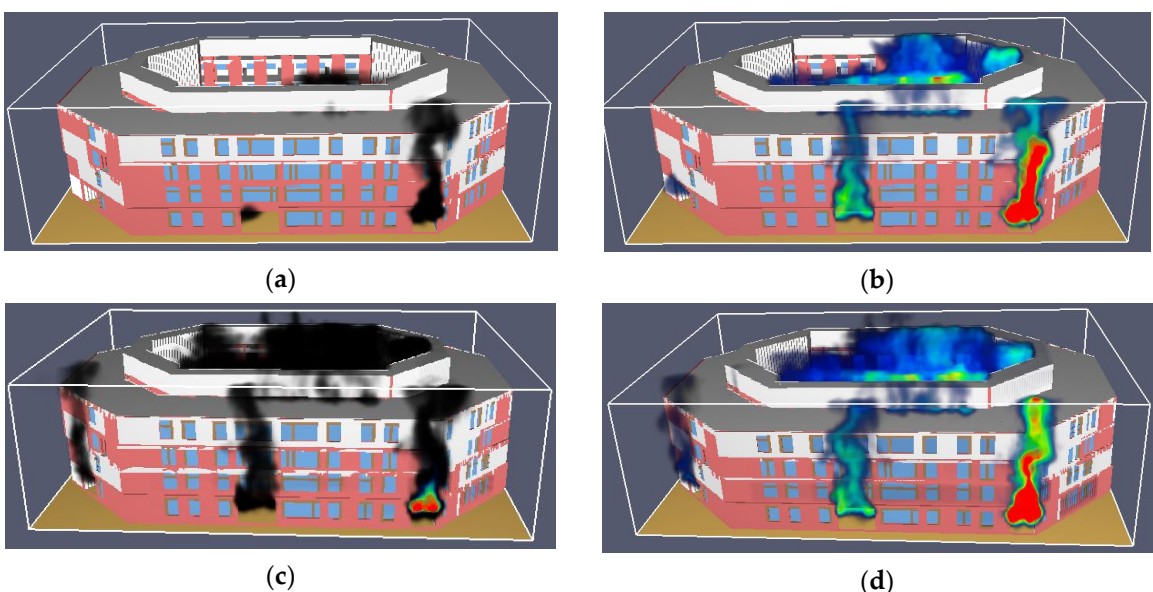

**Figure 8.** SmokeView simulation results for different combustion times: (**a**) 60 s; (**b**) 120 s; (**c**) 240 s; (**d**) 600 s.

## 5.3. Results

### 5.3.1. Impact of Path Accessibility on Indoor Routing

The path accessibility of a fire scene is determined by the state of the nodes, the number of flammable and explosive objects, and the number of non-flammable obstacles. These are important factors that affect the indoor evacuation path. Without considering the impact of other environmental semantics, we simulated and planned an indoor navigation evacuation path from the start node StartNode (f4_r17) to the first floor of the anchor space exit to analyze the impact of accessibility on the indoor fire emergency evacuation routing.

The semantic setting of the path accessibility environment of the fire scene is shown in Table 5. The ideal path planning situation in the simulation of all areas in scenario 1 is not on fire, and the obstacles in the path that hinder the passageway are as follows. In scenario 2, we simulated path planning when a fire occurs with obstacles. The experimental results are shown in Figure 9. The results of the indoor routing algorithm for the fire scenes are shown in black lines in Figure 9a in the absence of a fire in scenario 1. The sequence of nodes passed is f4_r17 → f4_w20 → f4_c43 → f4_c41 → f4_e1 → f3_e1 → f2_e1 → f1_e1 → f1_d46. In scenario 2, it was assumed that a fire occurs in the building, the door node f4_d22 in room A414 corresponding to f4_r17 is locked (red node in Figure 9b), and there are flammable and explosive objects from f1_c34 to f1_d46 (red section in Figure 9b). The results of the path planning in this case are shown as black lines in Figure 9b. The order of the nodes passed is f4_r17 → f4_d22 → f4_c42 → f4_c41 → f4_c40 → f4_d47 → f4_s07 → f3_s08 → f3_s07 → f2_s08 → f2_s07 → f1_s08 → f1_s07 → f1_d45 → f1_c33 → f1_d28 → f1_r16 → f1_w14. A comparison of the planned routes for scenarios 1 and 2 shows that when a fire does not occur, the elevator acts as a transit node and its state is reachable. At this time, the user can directly reach the first-floor exit via

the elevator. Once a fire occurs, the elevator nodes are unreachable for safety concerns. Stairs are used as the vertical channel for evacuation. The planned path of the algorithm also effectively avoids unreachable nodes and flammable and explosive objects. This is because $f_{acc} \to \infty$ when there are unreachable nodes and flammable and explosive objects according to the indoor 3D navigation traffic cost function $G(n)$ proposed in this paper. When planning the shortest path using the A* algorithm, $F(n) \to \infty$. This effectively avoids unreachable nodes and flammable and explosive objects and selects the f1_w24 node (window of A110 classroom) with a smaller evaluation function as the final exit node in the multi-anchor space exit list. Indoor path finding that considers accessibility in a fire scenario is therefore achieved.

**Table 5.** Environmental semantics of experiment 1.

| Scenario Number | Fire | Start Node | Unreachable Node | Obstacle |
|---|---|---|---|---|
| 1 | Yes | f4_r17 | No | No |
| 2 | No | f4_r17 | f4_d22 | f1_c34 to f1_d46 |

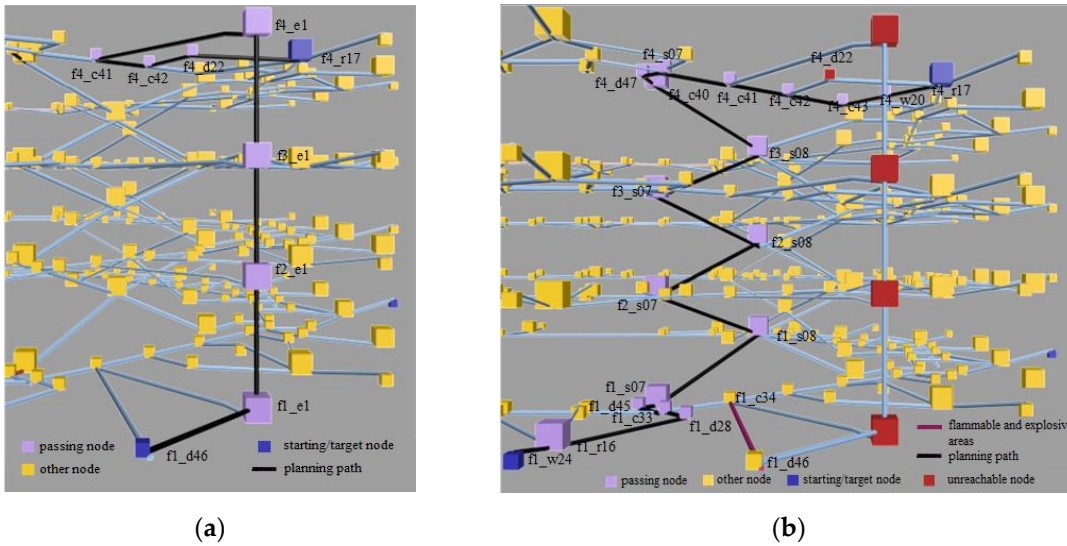

(**a**) (**b**)

**Figure 9.** Impact of path accessibility on indoor pathfinding. (**a**) Planning path in scenario 1; (**b**) Planning path in scenario 2.

## 5.3.2. Impact of Path Recognition Degree on Indoor Routing

The path recognition degree of a fire scenario is an important factor that affects safety and efficiency when escaping from a fire. Without considering the impact of other indoor semantics, we simulated the indoor navigation path from StartNode (f4_r17) to the first-floor anchor space exit and analyze the impact of accessibility on routing in the indoor fire scenarios.

In the case of an indoor fire, the path recognition degree mainly depends on the number of effective emergency evacuation indicators. The semantic setting of the simulation scenario is shown in Table 6. Assuming that there were indoor fires in both scenarios, scenario 1 simulated the ideal path planning situation when the emergency evacuation indicators in all areas are in good condition. Scenario 2 simulated the path selection when some emergency evacuation indicators fail. The experimental results are shown in Figure 10. The route planned by the routing algorithm in scenario 1 is identified by the black route in Figure 10a, and the node sequence is f4_r14 → f4_d26 → f4_c32 → f4_c31 → f4_c30 → f4_c29 → f4_d40 → f4_s01 → f3_ s02 → f3_s01 → f2_s02 → f2_s01 → f1_s02 → f1_s01 → f1_c34 → f1_d46 → f1_r13 → f1_w31. In scenario 2, the emergency evacuation indicators in the corridor on the fourth floor is set to fail. The results of the path planning in this case are shown as black lines in Figure 10b. The node sequence is as follows: f4_r14 → f4_w29 → f4_c33 → f4_c34 → f4_c35 → f4_c36 → f4_c37 → f4_c38 → f4_c39 → f4_c40 → f4_d47 → f4_s07 → f3_s08 → f3_s07 → f2_s08 →

f2_s07 → f1_s08 → f1_s07 → f1_d45 → f1_c33 → f1_c34 → f1_d46. Figure 10 shows that the planned path changes significantly. Because the failure condition of the emergency indicator lamp was not considered in scenario 1, the path recognition degree of each area in the building is consistently high and the planned route obtained using the routing algorithm for fire scenes cannot be well applied to the actual situation. In scenario 2, corridor node f4_ c27 on the west side of the fourth floor to node f4_ c34 emergency evacuation indicator is invalid. In this case, the fire smoke greatly reduces the visibility inside the building. The visibility of nodes f4_ c28 to f4_ c34 is less than 2 m, which makes users unable to quickly identify the direction without the guidance of emergency evacuation-indicating lighting, which reduces the escape efficiency. According to the indoor 3D navigation traffic cost function $G(n)$ proposed here, when there are nodes that are difficult to recognize, $f_{acc} \to \infty$. When planning the shortest path using the A* algorithm, $F(n) \to \infty$. This effectively avoids the difficulty of recognizing nodes. The f1_w24 node (window of A110 classroom) with a smaller evaluation function is selected as the final exit node in the multi-anchor space exit list.

**Table 6.** Environmental semantics of experiment 2.

| Scenario Number | Fire | Start Node | Emergency Evacuation Indicating Lamp |
|---|---|---|---|
| 1 | Yes | f4_r17 | No |
| 2 | Yes | f4_r17 | f4_c27 to f4_c34 |

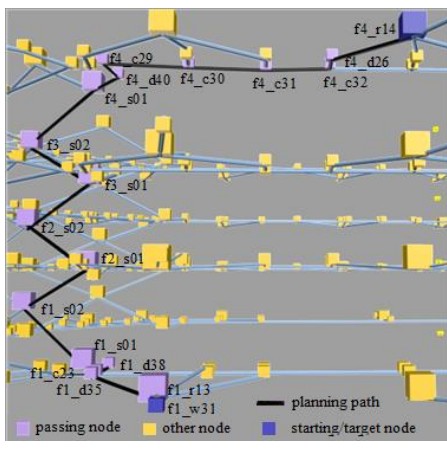

**(a)**

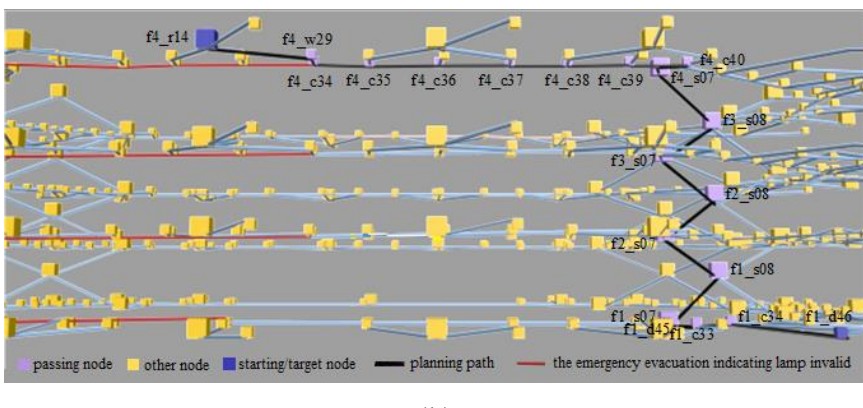

**(b)**

**Figure 10.** Impact of path recognition degree on indoor pathfinding. (**a**) Planning path in scenario 1; (**b**) Planning path in scenario 2.

### 5.3.3. Impact of Fire Parameters on Indoor Path Planning

Fire parameters have a critical impact on human health threats and emergency evacuation in a fire. We simulated two situations to illustrate the effect of fire parameters on indoor routing without considering the interference of other environmental factors. An indoor escape path was planned starting from node f4_r17. The simulated environment semantic settings are shown in Table 7. Because fire parameters are related to time, it is assumed that people perceive and start to escape after 30 s of fire with an escape speed of 1.3 m/s. The time for people to reach the adjacent node is estimated by calculating the ratio of the distance from the current node to the adjacent node at this speed. By querying the FDS simulation results, we calculated and updated the A* algorithm evaluation function values. In scenario 1, there is no fire and the influence of fire parameters is not considered. The planned path is shown as the black route in Figure 11a: f4_r17 → f4_d22 → f4_c42 → f4_c41 → f4_c40 → f4_d47 → f4_s07 → f3_s08 → f3_s07 → f2_s08 → f2_s07 → f1_s08 → f1_s07 → f1_c33 → f1_c34 → f1_d46. In scenario 2, the initial room temperature was set to 20 °C. In the event of a fire, the fire parameters changed with time and affected the environmental conditions. The path planning results at this time are shown in Figure 11b. The red triangle indicates the area where the fire source is located in the simulation. The node sequence is as follows: f4_r17 → f4_d21 → f4_c45 → f4_c46 → f4_c47 → f4_c48 → f4_d44 → f4_s10 → f3_s11 → f3_s10 → f2_s11 → f2_s10 → f1_s11 → f1_s10 → f1_d42 → f1_c42 → f1_r22 → f1_w04. Because the influence of fire parameters was not considered in scenario 1, the algorithm uses f4_r17 as the starting node according to the principle of the A* algorithm and calculates the path to the lowest outdoor travel cost through the Euclidean distance estimation function.

**Table 7.** Environmental semantics of experiment 3.

| Scenario Number | Fire | Start Node | Fire Parameters |
|:---:|:---:|:---:|:---:|
| 1 | Yes | f4_r17 | No |
| 2 | Yes | f4_r17 | f4_c27 to f4_c34 |

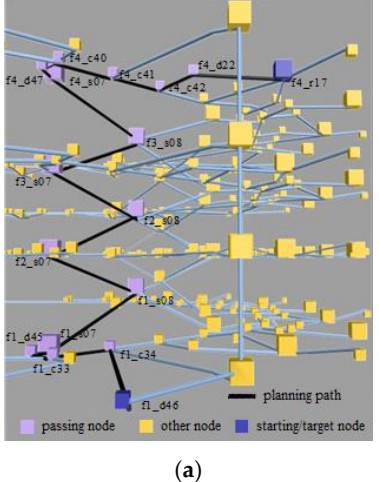
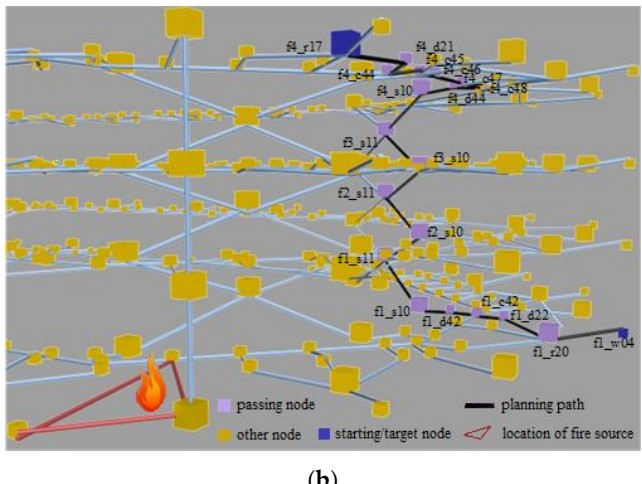

(**a**)　　　　　　　　　　　　　　　　　　　　　　　　　　(**b**)

**Figure 11.** Impact of fire parameters on emergency evacuation routing. (**a**) Planning path in scenario 1; (**b**) Planning path in scenario 2.

The result of the path planning does not consider the threats to human safety caused by fire temperature and smoke visibility. In the obtained planned path, node f1_s07 has a temperature higher than 100 °C and a visibility of less than 0.5 m after 150 s of fire. Node f1_d46 has a temperature of higher than 130 °C and a visibility of less than 0.3 m after 200 s. According to the calculated escape speed, before node f1_s07 can be reached, the temperature will cause serious harm and the escaping people cannot pass through this node. As shown in Table 2, when the human body temperature exceeds 42 °C, central nervous system functioning is disrupted and the proteins in the body may be

denatured and solidified, which is life threatening. Node f1_d46 is selected as the target node in the planned path and its temperature far exceeds the limit that the human body can withstand. In the case of an indoor fire, the path planning results directly obtained using the classic A* algorithm are unreliable. In contrast, when we calculated the path considering the influence of fire parameters in scenario 2, the evaluation function was calculated according to the indoor fire parameter cost function proposed here. The resulting path is that with the lowest travel cost when considering the influence of factors, such as fire temperature and smoke visibility. The temperatures of all of the nodes in the path do not exceed 30 °C, and nodes such as f1_s07 and f1_d46 that could threaten safety are avoided.

A comparison of the path results in scenarios 1 and 2 shows that for indoor routing with different fire parameters, the algorithm avoids dangerous road sections by sensing the fire environment information and obtains different path results. This is because the algorithm considers different fire parameters and updates the weight of the navigation traffic cost function according to the fire parameter levels. Higher fire parameters are associated with more dangerous routes and higher navigation costs. The escape route for an emergency evacuation can therefore be changed to achieve a reasonable planning path.

### 5.3.4. Indoor Routing Constrained by Multi-Semantic Parameters

We considered all of the influential factors (path accessibility, path recognition degree, and fire parameters), and set the start node to f4_r16 to simulate the emergency evacuation path with different semantic parameters to verify the effectiveness of the proposed method. We jointly considered path accessibility, path recognition degree, and fire parameters in this experiment.

Scenario 1 was assumed to simulate an ideal path planning situation that is not affected by fire parameters: the path is clear, and the emergency evacuation indicators are valid. Scenario 2 was a fire situation with non-flammable obstacles. The emergency evacuation indicators in the red area on the fourth floor fail, and the evacuation path is simulated. The results are shown in Figure 12. In scenario 1, the indoor path planning is shown by the black route in Figure 12a. The path is f4_r16 → f4_d30 → f4_c40 → f4_c41 → f4_e1 → f3_e1 → f2_e1 → f1_e1 → f1_d46. In Scenario 2, nodes f4_e1, f3_e1, f2_e1, and f1_e1 were set as unreachable and the emergency evacuation indicators in the red area on the fourth floor were set as invalid. There are also a large number of non-flammable obstacles in the room nodes. The planning path obtained by the indoor routing algorithm for this scene is shown in Figure 12b, and the path is f4_r16 → f4_d29 → f4_c38 → f4_c37 → f4_c36 → f4_c35 → f4_c33 → f4_c34 → f4_c32 → f4_c31 → f4_c30 → f4_c29 → f4_d40 → f4_s01 → f3_s02 → f3_s01 → f2_s02 → f2_s01 → f1_s02 → f1_s03 → f1_d39 → f1_c17 → f1_c16 → f1_w01 → f1_r07 → f1_w02. Figure 12a,b show that the paths are planned quite differently. Because scenario 1 is not affected by fire, the user directly reaches the first-floor exit through the elevator node to evacuate. In scenario 2, due to the fire and given the influence of path accessibility, path recognition degree, and fire parameters, the state of the elevator node is unreachable, and people cannot use the elevator as an escape route. Due to the failure of the emergency evacuation indicators at the fourth floor, people evacuate to the stairs node on the third floor to escape. At this time, because of the fire temperature and smoke visibility, people leave this stair assembly, use another stair assembly to evacuate, and finally reach the anchor space exit node f1_w02. If the escape route planned in scenario 1 is used in the event of a fire, exit node f1_d46 is greatly affected by the fire and the temperature will reach more than 100 °C after 150 s; it cannot therefore be used to successfully evacuate people. In scenario 2, the planning path algorithm fully considers the fire influencing factors, avoids nodes that are difficult to recognize, unreachable, or greatly affected by the fire parameters, and successfully chooses the first-floor window component f1_w02 as the escape exit. The experimental results show that the proposed method effectively senses multi-semantic changes in the case of a fire and plans a safe and effective indoor fire emergency evacuation path.

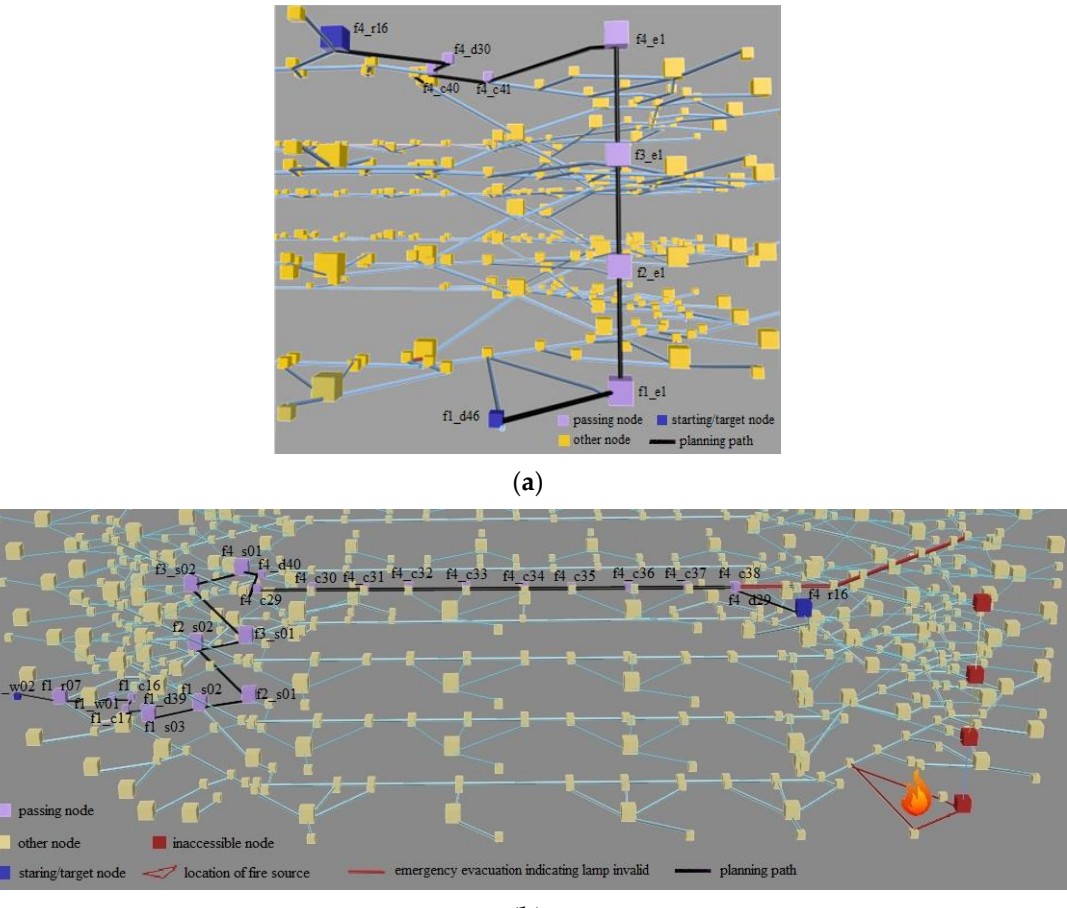

**Figure 12.** Impact of multi-semantic constraints on indoor 3D pathfinding. (**a**) Planning path in scenario 1; (**b**) Planning path in scenario 2.

## 6. Conclusions

We propose a multi-semantic indoor three-dimensional emergency evacuation routing method that comprehensively considers path accessibility, path recognition degree, and fire parameters. The results show that the method achieves the dynamic perception of the semantic changes of an indoor fire by accounting for the navigation traffic cost function for a fire scenario, and avoids path sections that are unreachable, difficult to recognize, and/or greatly affected by temperature and visibility. This method provides users with easy and safe path planning results. As a next step, we will further consider the impact of firefighters on emergency evacuation and implement a two-way routing algorithm for evacuation and rescue.

**Author Contributions:** Conceptualization, Yan Zhou; data curation, Yuling Pang; formal analysis, Yan Zhou; investigation, Yuling Pang; methodology, Yan Zhou and Yuling Pang; project administration, Yan Zhou, Fen Chen, and Yeting Zhang; supervision, Yan Zhou and Yeting Zhang; validation, Yuling Pang; Writing—Original draft, Yuling Pang; and Writing—Review and editing, Yan Zhou, Yuling Pang, and Fen Chen. All authors have read and agreed to the published version of the manuscript.

**Funding:** This work was partly supported by the National Key Research and Development Program of China (No. 2018YFB0505501 and 2016YFB0502303), and National Natural Science Foundation of China (No. 41871321, 41471332 and 41871314).

**Conflicts of Interest:** The authors declare no conflict of interest.

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
