# Peer review of "Three-Dimensional Indoor Fire Evacuation Routing"

_ijgi, doi:10.3390/ijgi9100558_

Round 1

Reviewer 1 Report

In this paper, the authors develop a multi-semantic constrained indoor three-dimensional (3D) fire evacuation routing method that considers multi-dimensional indoor fire scene-related semantics, such as path accessibility, path recognition degree, and fire parameters. The results are analyzed and compared in simulation experiments.

The work is extensively presented.

Abstract: clear

Introduction/related works: it is a bit contrasting that in the introduction the authors state that such work hasn’t really been done and that they then give an overview of the work done in the “related work”-section. In the last paragraph, the authors state then the real difference between these works and their work. This has to be restructured a bit in order to make things clearer.

The Three-dimensional expression model for indoor fire scenarios is clearly and extensively presented and explained, and extensions to current existing models are discussed.

Also the 3D routing method for indoor fie scenarios is clearly explained and presented.

The presented method is extensively exemplified by an illustrative example.

Overall, this is interesting work.

Author Response

Point 1:

Introduction/related works: it is a bit contrasting that in the introduction the authors state that such work hasn’t really been done and that they then give an overview of the work done in the “related work”-section. In the last paragraph, the authors state then the real difference between these works and their work. This has to be restructured a bit in order to make things clearer.

Response 1: Thank you for your good suggestion. These two parts have been restructured.

Reviewer 2 Report

This paper presents a 3D indoor data model for fire emergency response and a route computation algorithm. It also shows an experiment conducted with real site. The topics are interesting, however serious weakness and issues are found;

-[22] looks nothing to do with IndoorLocationGML. And no literature is given on it.

- The parts for requirement analysis are excellent.

- Many concepts of this paper come from IndoorGML. However no clear description about how they are related and borrow from IndoorGML.

- Actually IndoorGML provides a sound basis for indoor space modeling from which most application models can be extended. For example the feature types presented in Figure 1 include many classed defined in IndoorGML and the remaining ones can be derived from those in IndoorGML.

- Many concepts are taken from IndoorGML without any citation. CellSpace and CellSpaceBoundary are among main classes of IndoorGML core module however no citation is found. Figure 3 and its explanation are from a paper in IJGI (doi:10.3390/ijgi6040116) without any citation. This may be considered as a plagiarism.

- The UML notations in Figure 2 are not correctly used. For example window and door are subclasses of opening (not its components) and similarly room and channel are subclass of CellSpace, while they are modeled as components.

- Notations in eq.6-8 are strange. How to distinguish two different functions by only its parameters? And what happens when f_state(v_i)=0? The authors try to show the infinite but 

- Section 4 is straightforward. But due to the dynamic properties of indoor fire, a path found by A* could be blocked by expansion of smoke or fire without any escape. How can we guarantee the correctness of the solution in case of dynamic situation?

- I don’t see any difference between multi-level 2D routing and 3D routing. In order to run FDS, we need 3D model but it is not part of routing. Vertical connections can be also modeled by multi-level 2D indoor model. What is the point that authors try to claim by 3D routing?

- Actually subspacing is a critical factor for correct routing whether or not fire emergency. When we look at Figure 6 and its explanation, it is not clear how the granularity of subspace is made and how long hallways are divided.

Author Response

Point 1: [22] looks nothing to do with IndoorLocationGML. And no literature is given on it.

Response 1: Thank you very much for your careful reading. We have corrected the reference.

Point 2: Many concepts of this paper come from IndoorGML. However no clear description about how they are related and borrow from IndoorGML.

Response 2: The description has been added in line 189-195.

Point 3: Actually IndoorGML provides a sound basis for indoor space modeling from which most application models can be extended. For example the feature types presented in Figure 1 include many classed defined in IndoorGML and the remaining ones can be derived from those in IndoorGML.

Response 3: We agree with you. Our indoor model is also based on IndoorGML, which extends fire-related components for indoor fire evacuation, and integrates location semantics of IndoorLocationGML into the indoor model.

Point 4: Many concepts are taken from IndoorGML without any citation. CellSpace and CellSpaceBoundary are among main classes of IndoorGML core module however no citation is found. Figure 3 and its explanation are from a paper in IJGI (doi:10.3390/ijgi6040116) without any citation. This may be considered as a plagiarism.

Response 4: Thank you very much for your careful reading. The IndoorGML standard has been cited. We also feel really sorry for this big mistake in Fig. 3.  We have added this paper in the reference list and cited it in the paper.

Point 5: The UML notations in Figure 2 are not correctly used. For example window and door are subclasses of opening (not its components) and similarly room and channel are subclass of CellSpace, while they are modeled as components.

Response 5: We are sorry for our carelessness. We have corrected it and we also feel great thanks for your point out.

Point 6: Notations in eq.6-8 are strange. How to distinguish two different functions by only its parameters? And what happens when f_state(v_i)=0? The authors try to show the infinite but

Response 6: Eq. 7 and 8 are the same functions. The different parameters vi and vj denote the accessibility state of different nodes. When f_state(v_i)=0, the cost function of the path accessibility from node vi to node vj is infinite.

Point 7: Section 4 is straightforward. But due to the dynamic properties of indoor fire, a path found by A* could be blocked by expansion of smoke or fire without any escape. How can we guarantee the correctness of the solution in case of dynamic situation?

Response 7: A* algorithm is a heuristic algorithm, which find the path with the smallest cost according to the evaluation function F(n). F(n) consists of two parts, G(n) and H(n).  The G(n) relys on the fire parameters and indoor environment. When the changing of fire scenario happens, we update the value of G(n) in step 3 in our algorithm per call to A* to find a new path. If the blocked location lies in a key navigation node (i.e. the worst situation), the A* will fail to find a path.

Point 8: I don’t see any difference between multi-level 2D routing and 3D routing. In order to run FDS, we need 3D model but it is not part of routing. Vertical connections can be also modeled by multi-level 2D indoor model. What is the point that authors try to claim by 3D routing?

Response 8: We agree with you. It would be more understandable and proper to change “indoor 3D routing” into “indoor routing”.

Point 9: Actually subspacing is a critical factor for correct routing whether or not fire emergency. When we look at Figure 6 and its explanation, it is not clear how the granularity of subspace is made and how long hallways are divided.

Response 9: Thank you for your comments. We modified the Figure 6, which shows a subdivision of single-layer indoor space and provides navigable subspaces based on meshes.

Reviewer 3 Report

Interesting paper on indoor 3D routing for a fire scenario.

The Introduction and Related Works section needs to be updated with some supporting literature and some statements needs more context.

This reviewer suggest to add and discuss the following paper, recently published in ISPRS IJGI

https://www.mdpi.com/2220-9964/8/3/126

https://www.mdpi.com/2220-9964/6/12/384

line 101-103: "The OGC therefore specifically proposed IndoorGML [20] for indoor spaces, which is classified according to functions. However, IndoorGML lacks semantic descriptions for indoor positioning and navigation, and cannot fully meet the needs of indoor location services." This is some statement to make ... Does IndoorGML really lack semantic descriptions for indoor positioning and navigation? Could it not be combined with another semantic rich model like LADM, described in: https://www.int-arch-photogramm-remote-sens-spatial-inf-sci.net/XLII-4/11/2018/

line 107-108: "However, the amount of BIM data is too large to meet the needs for rapid response to path finding in indoor fire scenarios." Again, quite a statement to make ... Do you have some references to proof this statement? 

line 110: IndoorLocationGML: See previous remark on IndoorGML, check "TOWARDS THE INTEGRATION OF INDOORGML AND INDOORLOCATIONGML FOR INDOOR APPLICATIONS": https://www.isprs-ann-photogramm-remote-sens-spatial-inf-sci.net/IV-2-W4/343/2017/ 

References:

Have to be double checked, see for example: [3] and [24]

Author Response

Point 1: The Introduction and Related Works section needs to be updated with some supporting literature and some statements needs more context.

This reviewer suggests to add and discuss the following paper, recently published in ISPRS IJGI

https://www.mdpi.com/2220-9964/8/3/126

https://www.mdpi.com/2220-9964/6/12/384

Response 1: Thank you for pointing out these related works. We have cited them in the revision.

Point 2: line 101-103: "The OGC therefore specifically proposed IndoorGML [20] for indoor spaces, which is classified according to functions. However, IndoorGML lacks semantic descriptions for indoor positioning and navigation, and cannot fully meet the needs of indoor location services." This is some statement to make ... Does IndoorGML really lack semantic descriptions for indoor positioning and navigation? Could it not be combined with another semantic rich model like LADM, described in: https://www.int-arch-photogramm-remote-sens-spatial-inf-sci.net/XLII-4/11/2018/

Response 2: We have modified this part in order to make it clearer. Navigation inside buildings is performed to meet different indoor navigation’s desires. IndoorGML provides a general framework and semantic descriptions for indoor navigation. For indoor navigation in fire scenarios, we need to combine it with another semantic rich model as you pointed out. In our paper, we combined IndoorGML with the Chinese national standard of indoor navigation (IndoorLocationGML), and extended the indoor fire-oriented components based on the framework of the two standards in order to support the navigation requirement in fire situation. We agree with you that LADM is a semantic rich model, which assigns rights, restrictions and responsibilities to each indoor space and indicates the accessible spaces for each type of user. The combined LADM-IndoorGML model was used for indoor navigation by Alattas et al.. We have added this work in section 2. As the access rights of the indoor spaces are affected by the crisis event, LADM needs be modelled explicitly before crisis situation, and our building data need be also converted into the model. This is interesting but is a challenging work. We cannot finish this work in the limit time for the revision (10 day allowed by the associated editor) Thanks for your suggestion. We leave it as our future work.

Point 3: line 107-108: "However, the amount of BIM data is too large to meet the needs for rapid response to path finding in indoor fire scenarios." Again, quite a statement to make ... Do you have some references to proof this statement?

Response 3: We have modified this statement and provided related references in Section 3.1

Point 4: line 110: IndoorLocationGML: See previous remark on IndoorGML, check "TOWARDS THE INTEGRATION OF INDOORGML AND INDOORLOCATIONGML FOR INDOOR APPLICATIONS": https://www.isprs-ann-photogramm-remote-sens-spatial-inf-sci.net/IV-2-W4/343/2017/

Response 4: We have cited and discussed this paper according to your suggestion in Section 3.1.

Point 5: References: Have to be double checked, see for example: [3] and [24]

Response 5: Thanks for your careful reading. References have been checked and updated.